# Taking a Step Back:
# Revisiting Classical Approaches for Efficient Interactive Segmentation of Medical Images

*Zdravko Marinov[1,2] ⓘ, Alexander Jaus[1,2] ⓘ, Jens Kleesiek[3,4] ⓘ, and Rainer Stiefelhagen[1] ⓘ

[1] Karlsruhe Institute of Technology, Karlsruhe, Germany
[2] HIDSS4Health - Helmholtz Information and Data Science School for Health, Karlsruhe/Heidelberg, Germany
[3] Institute for AI in Medicine, University Hospital Essen, Essen, Germany
[4] Cancer Research Center Cologne Essen (CCCE), University Medicine Essen, Essen,
*Corresponding Author: zdravko.marinov@kit.edu

**Abstract.** Interactive segmentation plays a pivotal role in medical image analysis for several reasons. It enables clinicians to precisely delineate regions of interest for accurate diagnosis and treatment planning while also allowing for real-time interaction with rapid annotations without workflow interruptions. While the emergence of MedSAM in 2023 presented a promising solution with its modality-agnostic model, its efficiency is hindered by its large size, resulting in long inference times. In response, we revisited simpler models such as thresholding, k-means clustering, and shape-based slice interpolation for efficient interactive segmentation tailored to specific modalities. Surprisingly, these rudimentary expert models outperformed MedSAM in terms of both segmentation performance and computational efficiency on multiple imaging modalities reaching a Dice score of 85.65 and a Normalized Surface Dice of 86.68 on the validation set. Our findings show the need to compare to older, simpler approaches to unveil the limitations of emerging foundation models. By examining these approaches, we aim to discover why MedSAM fails on certain modalities and enhance its robustness and efficiency leading to a more reliable general model for the segmentation of medical images.

## 1 Introduction

**Background.** Advancements in deep learning have propelled the segmentation of anatomical structures and lesions in medical images. However, they often rely on manually annotated datasets [7,33,3,19,44,12]. Additionally, the volumetric nature of some imaging modalities such as CT, MRI, or PET, poses a significant challenge, as annotating each voxel demands extensive time and expertise. To circumvent this hurdle, interactive segmentation methodologies have emerged, utilizing less labor-intensive annotations such as clicks or bounding boxes, rather than dense voxelwise labels [31,48,9,10,4,43,26,42,32]. These interactive models

integrate user interactions with the image input to generate predictions, guided by these interactions. Once validated by medical experts, these predictions can serve as new labels [31].

**Related work.** Recently, MedSAM [28] released a fine-tuned Segment Anything Model (SAM) [24] on 11 imaging modalities and over 1.5 million image-mask pairs. MedSAM demonstrates great generalizability across various imaging modalities and segmentation tasks [28,31]. However, its large size hinders the real-time interaction between the annotator and the model. There have been multiple light-weight versions of MedSAM such as MobileSam [46] and EfficientViT-SAM [47] that optimize the efficiency of MedSAM while retaining most of its generalization to multiple modalities. To explore this further, Ma et al. [28] hosted the Segment Anything In Medical Images On Laptop Challenge[5] to gather insights on how to design efficient bounding-box-based methods for interactive segmentation. This paper describes our submission to this challenge.

**Motivation.** We aim to revisit classical methods as they offer a simple and efficient solution to most segmentation tasks. By comparing them to current generalist models such as MedSAM, we aim to gain insights into how such simple models can outperform large pre-trained vision models and delve into the discussion of how to improve MedSAM in future iterations. Our work presents the following contributions:

1. We investigate classical approaches for 11 imaging modalities and investigate if they can outperform MedSAM's lightweight implementation (LiteMed-SAM[6]) in terms of segmentation accuracy and efficiency
2. We examine the failure cases and discuss why MedSAM silently fails on certain modalities and propose how to tackle this in future fine-tuning iterations
3. We make all our code and trained models publicly available to the community

## 2   Method

We go over each of the 11 imaging modalities one-by-one and examine which classical approaches are able to outperform MedSAM and propose techniques to make MedSAM more efficient on modalities on which we could not outperform it. The 11 modalities are: (1) Computed Tomography (CT); (2) Magnetic Resonance Imaging (MRI); (3) Positron Emission Tomography (PET); (4) Ultrasound (US); (5) Dermoscopy; (6) Microscopy; (7) Mammography; (8) X-Ray; (9) Endoscopy; (10) Fundus; and (11) Optical Coherence Tomography (OCT).

***Note:*** We only focus on the segmentation tasks seen in the MedSAM training dataset, e.g., only FDG-PET lesions segmentation and only optic disc segmentation on fundus images. We also always use LiteMedSAM as a lightweight MedSAM implementation and refer to it as MedSAM for brevity.

---

[5] `https://www.codabench.org/competitions/1847/`
[6] `https://github.com/bowang-lab/MedSAM/tree/LiteMedSAM`

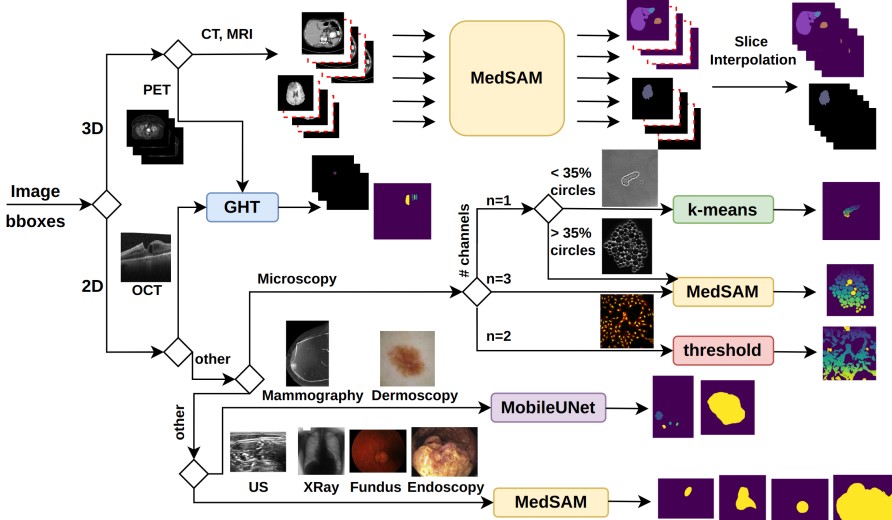

Fig. 1: Overview of our pipeline. We apply a different model for the various imaging modalities. For CT and MRI, we apply MedSAM [28] on a subset of all slices and interpolate the rest with shape-based interpolation [38]. For PET and OCT, we use the generalized histogram threshold (GHT) [5]. For microscopy, we check the number of channels and apply: (1) MedSAM [28] for $n = 3$; (2) a threshold, set to the mean image intensity for $n = 2$; (3) k-means clustering [30] for $n = 1$ if less than 35% of the bounding boxes contain circular objects, otherwise MedSAM. For mammography and dermoscopy, we train a lightweight MobileUNet [20], and for US, XRay, fundus, and endoscopy, we use MedSAM.

## 2.1   CT and MRI

**Tasks:** The main CT tasks seen in the MedSAM training dataset are extremely diverse as they focus on organs from all regions in the body as well as different diseases such as COVID-19 infections, various tumors, pleural effusion, etc. The same applies to the MRI targets consisting of brain tumors, abdominal organs, prostate cancer, and many other anatomical and pathological targets.

**Challenges:** The segmentation tasks are highly diverse, making it challenging and time-intensive to manually incorporate expert knowledge for all possible structures. Additionally, both CT and MRI images are 3-dimensional, requiring MedSAM to perform a forward pass for each slice, which results in significant computational overhead. Furthermore, the slice-wise predictions are computed independently, lacking a mechanism to ensure spatial consistency and smoothness across slices.

**Classical Approaches:** The diversity of potential targets in CT and MRI images necessitates a general model. Therefore, we did not replace MedSAM with a classical segmentation model. Instead, we reduced the number of predictions by

applying MedSAM to only a subset of the slices from the volume. The remaining slices are interpolated using shape-based interpolation as proposed in [38]. This interpolation is performed at the prediction level. Specifically, given two binary slice predictions obtained from MedSAM $p_i$ and $p_j$, with $j > i$, the intermediate slices $k \in \{i + 1, ..., j - 1\}$ are interpolated as follows:

$$p_k = \texttt{interp}(p_i, p_j, \frac{k - i}{j - i})$$

where $\texttt{interp}(\cdot)$ is defined in [38] and $\frac{k-i}{j-i} \in [0, 1]$ is the step size for the interpolation. To decide which slices to predict with MedSAM, we subsample the indices uniformly with a subsampling factor $s$. For example, $s = 2$ means we predict with MedSAM every second axial slice and interpolate the rest, $s = 3$ means we predict only every 3rd slice, etc. We always predict the first and last axial slice of the bounding box to avoid edge cases.

## 2.2  PET and OCT

**Tasks:** The PET data released in the challenge contains samples only from the AutoPET dataset [7] which focuses on the segmentation of whole-body active tumor lesions using Fludeoxyglucose (FDG) as a radioactive tracer. The OCT data also stems from only one dataset [1] and focuses on the segmentation of intraretinal cystoid fluid.

**Challenges:** There are very few public PET datasets for tumor segmentation [7,36,8] which makes it impossible to train large-scale foundation models on this modality. PET lesions in AutoPET are also with very small contrast to surrounding tissues and there are other healthy anatomical structures that also exhibit a large physiological uptake (heart, brain, bladder, etc.). Additionally, the best results from AutoPET 2023[7] are quite low (Dice Score of 0.36 in the first place) indicating that this task is far from trivial. Regarding OCT, the modality presents images with a high resolution and a very small target size of the retinal fluid voids, leading to a strong class imbalance.

**Classical Approaches:** Thresholding methods are popular in tumor segmentation from PET scans [22,17,34] and lead to promising results as they are quite simple and intuitive to use. Bounding-box approaches are particularly advantageous because thresholds can be applied within the local context of the box, effectively excluding healthy tissues like the heart and brain that lie outside the defined boundaries, in contrast to previous methods that explicitly remove the brain and bladder from the global context of the whole body [35,7,11]. The bounding box also alleviates the class imbalance in OCT images as it constrains the input to only the local context around the object instead of the global high-resolution image.

We compute the Generalized Histogram Threshold (GHT) [5] using the combined PET or OCT values from all bounding boxes. This threshold is then

---

[7] https://autopet-ii.grand-challenge.org/leaderboard/

applied to the entire volume to generate a single prediction, with instance indices assigned according to their respective bounding box indices. Additionally, we observe that 99% of the PET tumors occupy between 20% and 88% of their bounding box volume in the training set. To ensure consistency, we apply dilation and erosion to all predictions falling outside this interval until they conform to it. If thresholding results in an empty prediction, we place a foreground voxel within the bounding box and dilate it until it reaches at least 20% of the box volume. For OCT images, we perform the same procedure but do not enforce a target volume between 20% and 88%. For PET images we keep all values above the threshold as tumors are characterized by a high FDG uptake, whereas for OCT we keep all values below the threshold as the cystoid fluids are darker.

### 2.3  Ultrasound

**Tasks:** The ultrasound tasks in MedSAM's training data include: (1) breast cancer segmentation; and (2) fetal head segmentation.

 **Challenges:** The ultrasound domain presents several challenges due to weak boundaries and the diversity of tasks (e.g., thyroid, kidney, cardiac structures, fetal head, breast cancer). This necessitates a general model capable of performing well on unseen data, even when the specific task is unknown. MedSAM struggles with fetal head segmentation because the labels are always perfect ellipses, while MedSAM attempts to fit the exact contour present in the image. This label bias can be mitigated by introducing a "prediction bias," where the prediction is also a perfect ellipse. However, this approach requires prior knowledge that the task is fetal head segmentation, which is not available during test time.

 **Classical Approaches:** Although we experimented with ellipse-based template matching, this was not included in the final submission. Additionally, we trained a MobileUNet [20] on ultrasound images from the hc18 [13] and Breast-US [2] datasets. However, the results were suboptimal, as shown in Table 7 so we resorted to using MedSAM in our final submission.

### 2.4  Dermoscopy

**Tasks:** The dermoscopy tasks in MedSAM's training data are limited to a single dataset, ISIC 2018 [6], which focuses on skin lesion segmentation.

 **Challenges:** This domain is relatively simpler than others, as skin lesions typically have "blobby" shapes and prominent features. However, the dataset exhibits high variability in annotation styles, as shown in Fig. 2. Some lesions are annotated with detailed boundaries, while others are marked with only a few lines to indicate the lesion boundary.

 **Classical Approaches:** Although we did not employ any classical non-deep learning methods, we trained a MobileUNet [20] specifically on the ISIC 2018 dataset [6]. The training details are provided in Section 3.2.

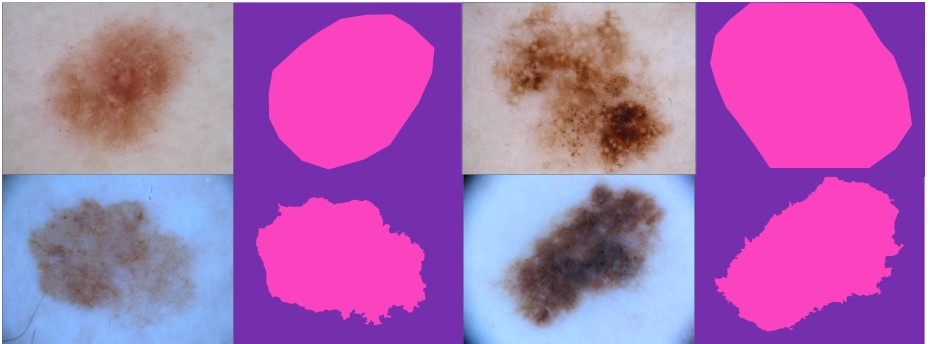

Fig. 2: Difference in annotation styles for skin lesions. Row 1: very coarse annotation with straight lines as boundaries. Row 2: fine-grained annotations with detailed lesion boundaries.

### 2.5   Microscopy

**Tasks:** MedSAM's microscopy training data is sourced solely from the NeurIPS 2022 CellSeg dataset [29]. However, this dataset presents a wide array of challenges, featuring diverse images captured using different microscope types, including brightfield, fluorescent, phase-contrast (PC), and differential interference contrast (DIC). Moreover, the dataset encompasses various cell types as segmentation targets, adding further complexity to the task.

   **Challenges:** The microscopy imaging modality presents several significant challenges: (1) The number of instances per image can be exceptionally high, exceeding 1000 in some cases, resulting in computational overhead when performing a forward pass for each bounding box. (2) The diversity of microscope types necessitates either a robust generalist model or multiple specialist models to accommodate various imaging characteristics. (3) The high-resolution nature of the images poses a challenge, as details may be lost when resizing to smaller resolutions, as is the case with MedSAM's resizing to $256 \times 256$.

   **Classical Approaches:** We adopt different classical methods depending on the number of channels present in the image.

   **Grayscale:** When dealing with grayscale images, we utilize a k-means clustering approach [30] with $k = 2$. To determine which class corresponds to the foreground, we compute the frequency of pixels belonging to each class within a $10 \times 10$ window at the center of the bounding box. We then select the class with the higher pixel count within this window. As depicted in Fig. 7, MedSAM encounters challenges with grayscale images where the target is dark and the bounding boxes are relatively small compared to the entire image. In contrast, k-means clustering achieves more plausible segmentations for such cases. However, we did not observe this problem with circular cells, so we opted to continue using MedSAM for cases where more than 35% of the bounding boxes contain a circular object. To detect circles, we employed the Hough circle transform [23]

with various radii to check for the presence of circle-like objects within each bounding box.

**Two-channel:** In the case of images containing only two channels (typically 3 channels in practice, with one filled with zeros, such as in fluorescent microscopy), we simply utilize the mean image intensity as a threshold. This approach is straightforward yet quite effective.

**RGB:** For RGB images, we opt to use MedSAM directly, as it demonstrates robust performance in such cases.

### 2.6   Mammography

**Tasks:** The training set comprises only one publicly available dataset: CDD-CESM [21], which addresses breast abnormalities such as calcifications and benign and malignant tumors.

**Challenges:** The target structures exhibit considerable variability in size, and the masks are annotated at a coarse level, similar to some labels observed in the ISIC 2018 dataset [6], as illustrated in Fig. 3.

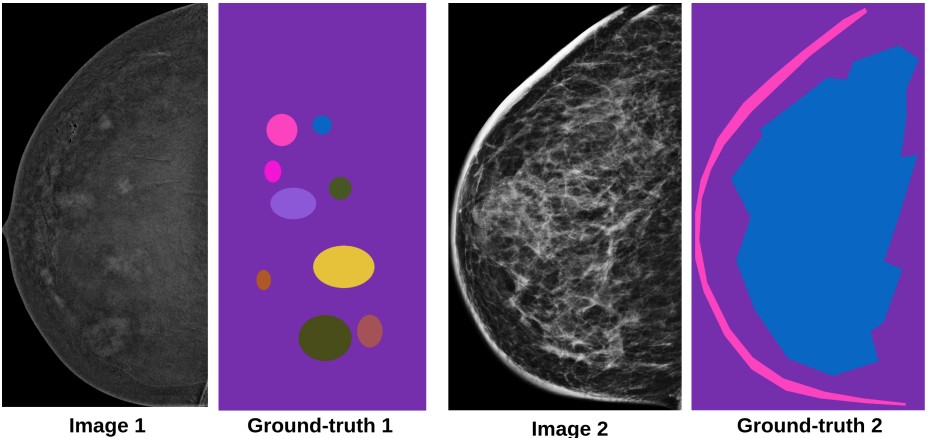

| Image 1 | Ground-truth 1 | Image 2 | Ground-truth 2 |

Fig. 3: Examples of images and their ground-truth masks from the training set. The labels are coarse and the targets vary strongly in size.

**Classical Approaches:** We did not employ any non-deep learning approaches for mammography but we did train a MobileUNet [20] to improve the inference speed.

### 2.7   X-Ray, Endoscopy, and Fundus

**Tasks:** The X-Ray, Endoscopy, and Fundus modalities encompass a range of tasks. X-Ray imaging primarily targets anatomical structures like lungs [15] and

specific pathologies such as COVID-19 lesions [41]. Endoscopy images are focused on detecting polyps [18] or surgical instruments [14]. In contrast, Fundus images in the training set concentrate solely on optic discs and cups [37]. However, for more intricate structures like vessels in fundus images, the bounding-box interaction signal proves inadequate for highlighting relevant context, and thus was not considered for this challenge.

**Challenges:** X-Ray images exhibit a strong diversity, featuring various structures. The validation set even includes dental X-Rays requiring teeth segmentation, which requires a model capable of generalizing to such unseen structures. Hence, we employed MedSAM for this modality. In contrast, Fundus and Endoscopy images offer a simpler and less diverse setting than X-Ray. Despite our efforts, we were unable to surpass MedSAM's performance in either of these domains, leading us to utilize it for our final submission for all three modalities.

**Classical Approaches:** We did not apply any classical approaches to these domains due to time constraints and utilize MedSAM for our submission.

Table 1: Summary of our used models for the final submission.

| Modality | Used Model |
|---|---|
| CT and MRI | LiteMedSAM [28] with slice interpolation $s = 3$ [39] |
| PET and OCT | Generalized Histrogram Threshold [5] |
| US, X-Ray, Fundus, Endoscopy | LiteMedSAM [28] |
| Mammography and Dermoscopy | MobileUNet [20] |
| Microscopy | k-means, thresholding, and LiteMedSAM [28] |

### 2.8   Preprocessing

We re-used the code provided by LiteMedSAM[8] for loading the data and inferring predictions and added more functions to the script for our methods. We avoid loading LiteMedSAM's weights for tasks which do not need it and import modules only immediately before they are used. The image loading and preprocessing is done as follows:

**LiteMedSAM:** The image is resized to a common size of $256 \times 256$ and padded to the shorter side to keep the original aspect ratio. Then, the image is min-max normalized and fed to the model. The model performs a forward pass for each bounding box.

**MobileUNet:** We iterate over each bounding box separately. We first crop the image according to the bounding box and then resize the crop to a common size of $256 \times 256$ and pad the shorter side to keep the original aspect ratio. Then, the crop is min-max normalized and fed to the model. The prediction is then resized to the original crop resolution and inserted in the final prediction.

---

[8] https://github.com/bowang-lab/MedSAM/blob/LiteMedSAM/CVPR24_LiteMedSAM_
infer.py

**k-means:** When applying k-means clustering, we use the unnormalized values within each bounding box and apply k-means for each bounding box.

**Thresholding:** Thresholds are always computed using the combination of all image values in the bounding boxes and then applied to the whole unnormalized image. Instance indices are then assigned according to the bounding box indices.

### 2.9   Post-processing

After each forward pass, we perform two post-processing transformations. For all 2D images, except the ones predicted by MedSAM, we keep the largest connected component and fill all the holes within it. The second transform, which we apply to all images, regardless of the imaging modality or used model, is to filter all the instance predictions that are outside of the bounding box.

## 3   Experiments

### 3.1   Dataset and evaluation measures

We used only the challenge dataset for model development and validation. The evaluation metrics include two accuracy measures—Dice Similarity Coefficient (DSC) and Normalized Surface Dice (NSD), alongside one efficiency measure: running time. These metrics collectively contribute to the ranking computation.

### 3.2   Implementation details

**Environment settings**   The development environments and requirements for all our methods (except the training of MobileUNet) are presented in Table 2.

Table 2: Development environments and requirements for all our methods except MobileUNet training.

| System | Ubuntu 22.04.4 LTS |
| --- | --- |
| CPU[9] | Intel(R) Core(TM) i7-13700H CPU@5.00GHz |
| RAM | 8×4GB; 5200MT/s |
| GPU (number and type) | None |
| CUDA version | 11.8 |
| Programming language | Python 3.10.14 |
| Deep learning framework | torch 2.2.1 |
| Specific dependencies | None |
| Code | `https://github.com/Zrrr1997/medsam_cvhci` |

The development environments and requirements for the training of Mobile-UNet are presented in Table 2.

---

[9] `https://ark.intel.com/content/www/us/en/ark/products/232128/intel-core-i7-13700h-processor-24m-cache-up-to-5-00-ghz.html`

Table 3: Development environments and requirements for MobileUNet training.

| System | Red Hat Enterprise Linux release 8.8 (Ootpa) |
|---|---|
| CPU | Intel(R) Xeon(R) Platinum 8368 CPU @ 2.40GHz |
| RAM | 502GB |
| GPU NVIDIA A100-SXM4 40GB | |
| CUDA version | 11.8 |
| Programming language | Python 3.10.14 |
| Deep learning framework | torch 2.2.2 |
| Specific dependencies | None |
| Code | `https://github.com/Zrrr1997/medsam_cvhci/` |

**Training protocols** For MedSAM, we use the provided pre-trained LiteMed-SAM model whose training is described in [28]. For MobileUNet, we trained a model for the dermoscopy and mammography modalities using the ISIC 2018 dataset [6] and the CDD-CESM dataset [21] respectively. In both cases we apply the same training protocol: We train for 500 epochs with a learning rate of 5e-5, and a batch size of 4. We use the Adam optimizer with $\beta_1 = 0.9, \beta_2 = 0.999$ and reduce the learning rate by 10% if the loss has not decreased in the last 5 epochs. As a loss function we use the summation between Dice loss and cross-entropy loss because compound loss functions have been proven to be robust in various medical image segmentation tasks [27]. In each iteration, we sample one random bounding box from each image in the batch and resize it to $256 \times 256$ so that all inputs have a uniform resolution. The crops then form the input batch to the model. We do not use any data augmentation and select the checkpoint from the last epoch for both the mammography and dermoscopy models.

Table 4: Training protocol for Dermoscopy and Mammography MobileUNet.

| Pre-trained Model | None |
|---|---|
| Batch size | 4 |
| Patch size | $256{\times}256{\times}3$ |
| Total epochs | 500 |
| Optimizer | Adam ($\beta_1 = 0.9, \beta_2 = 0.999$) |
| Initial learning rate (lr) | 5e-5 |
| Lr decay schedule | ReduceLROnPlateau[10] (factor=0.9, patience=5) |
| Training time | 8.4 hours (Dermoscopy), 6.8 hours (Mammography) |
| Loss function | Dice Loss + Binary Cross-Entropy Loss (equal weights) |
| Number of model parameters 41.22M[11] | |
| Number of flops | 1.45G[12] |
| $CO_2$eq | 2.6KG (Dermoscopy), 1.5KG (Mammography) Kg[13] |

## 4  Results and discussion

We discuss the results of the individual modalities one-by-one as we propose different models for the 11 imaging modalities.

### 4.1  CT and MRI

**Efficiency Strategies:** The slice interpolation improves the efficiency as it is faster than MedSAM's forward pass. A higher subsampling factor $s$ leads to a better efficiency but it also leads to a drop in performance as high-frequency details are smoothened out and interpolation artifacts may occur (see Fig. 5).

Table 5: Results on the validation stage of the challenge for CT and MR images. Our final submission in indicated in **bold**.

| Model | Dice | NSD | Time per Image |
|---|---|---|---|
| LiteMedSAM (CT, no subsampling) | 92.35 | 95.09 | 47.3s |
| LiteMedSAM (CT, s=2) | 91.88 | 94.35 | 28.1s |
| **LiteMedSAM (CT, s=3)** | 91.59 | 94.18 | 21.3s |
| LiteMedSAM (CT, s=6) | 90.54 | 93.14 | 15.0s |
| LiteMedSAM (CT, s=8) | 89.81 | 92.42 | 13.0s |
| LiteMedSAM (MR, no subsampling) | 89.93 | 94.02 | 29.8s |
| LiteMedSAM (MR, s=2) | 88.96 | 92.90 | 18.1s |
| **LiteMedSAM (MR, s=3)** | 88.34 | 92.57 | 13.9s |
| LiteMedSAM (MR, s=6) | 86.43 | 90.88 | 10.2s |
| LiteMedSAM (MR, s=8) | 84.99 | 89.58 | 9.0s |

Table 5 shows that for a smaller subsampling factor $s \leq 3$, the performance loss is quite small ($\approx 1\%$ Dice and NSD) but the efficiency boost is quite high (>19s per sample). However, for larger factors $s > 3$ the performance declines further and this is clearly illustrated by the rough patterns and artifacts in Fig. 5. To balance the performance and efficiency, we opted for $s = 3$ for our final submission.

**How to improve MedSAM on CT and MRI?** MedSAM shows a remarkable robustness in these two imaging modalities but exhibits a slow inference as it processes the volumes slice-by-slice. As MedSAM is intended as a foundation model, we believe it should remain in the 2D domain so that it can process other imaging modalities such as OCT, dermoscopy, etc. However, a slice interpolation strategy proves to be quite reliable, especially for smaller subsampling rates, without decreasing MedSAM's performance dramatically. We believe that

---

[10] https://pytorch.org/docs/stable/optim.html
[11] https://github.com/sksq96/pytorch-summary
[12] https://github.com/facebookresearch/fvcore
[13] https://github.com/lfwa/carbontracker/

exploring such strategies in more detail can be beneficial for MedSAM's efficiency on the CT and MRI domains as well as to impose a smoothness constraint among adjacent slices since its slicewise predictions are independent of each other.

## 4.2   PET and OCT

**Efficiency Strategies:** Thresholding eliminates the need for slice-wise forward passes as the threshold is applied directly on the whole volume in a single operation. Table 6 shows that the thresholding outperforms MedSAM on all metrics and is 30x faster on the validation set. For OCT, this effect is less pronounced but still leads to both a better performance and a much higher efficiency on the training dataset as there are no image samples in the validation set.

Table 6: Results on the validation stage of the challenge for PET and on the training dataset for OCT

| Model | Dice | NSD | Time per Image |
|---|---|---|---|
| LiteMedSAM (PET) | 55.23 | 29.29 | 3.33s |
| Thresholding (PET) | 66.80 | 49.42 | 0.14s |
| LiteMedSAM (OCT) | 79.02 | 82.33 | 0.86s |
| Thresholding (OCT) | 86.34 | 88.64 | 0.38s |

**How to improve MedSAM on PET and OCT?** The results are concerning as a simple threshold and morphological operators can outperform MedSAM on the PET task. We suppose that the reason is the lack of available data to improve MedSAM so that it can generalize better and extract meaningful features. PET data can also be enhanced with anatomical labels derived from paired CT scans [16,35] which was the winning approach of AutoPET 2023 as it injects expert knowledge regarding the affected anatomical regions as additional information to the model. The OCT domain can also be improved by incorporating additional anatomical labels, e.g., corresponding to the various retinal layers.

## 4.3   Ultrasound

**Efficiency Strategies:** We did not improve the efficiency on this task as we use MedSAM in our final submission as our experiments with MobileUNet were unsuccessful as seen in Table 7.

**How to improve MedSAM on Ultrasound?** We observed that the fetal head labels in the training data can be effectively approximated by simple ellipses, as illustrated in Fig. 4, resulting in improved Dice scores and efficiency compared to MedSAM. However, ultrasound imaging is used for various other tasks, such as cardiac structure analysis [25], thyroid cancer detection [49], breast cancer assessment [2], and kidney imaging [40]. Therefore, relying solely on simple ellipses is not viable for all cases. One potential solution is to train a classifier

Table 7: Results on the validation stage of the challenge for Ultrasound images

| Model | Dice | NSD | Time per Image |
|---|---|---|---|
| LiteMedSAM | 94.78 | 96.81 | 0.66s |
| MobileUNet | 79.12 | 84.55 | 0.22s |

to identify when the ultrasound task involves fetal head segmentation. However, we were unsuccessful in achieving satisfactory results with this approach.

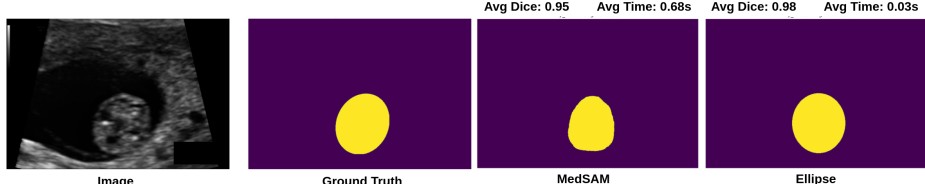

Fig. 4: Prediction for the hc18 dataset [13] with MedSAM and with an ellipse.

### 4.4  Dermoscopy

**Efficiency Strategies:** MobileUNet's forward pass is twice as fast as Med-SAM's, yet it maintains comparable performance on the validation set, as demonstrated in Table 8.

Table 8: Results on the validation stage of the challenge for Dermoscopy images

| Model | Dice | NSD | Time per Image |
|---|---|---|---|
| LiteMedSAM | 92.47 | 93.85 | 0.71s |
| MobileUNet | 92.63 | 94.22 | 0.31s |

**How to improve MedSAM on Dermoscopy?** We cannot make a statement on how to improve MedSAM in this domain as we have not been able to outperform it by a large margin.

### 4.5  Microscopy

**Efficiency Strategies:** The k-means clustering and the mean threshold are more than x10 faster than the MedSAM forward pass.

**How to improve MedSAM on Microscopy?** It seems that MedSAM struggles with miniature bounding boxes, especially since they are resized to

Table 9: Results on the validation stage of the challenge for Microscopy images

| Model | Dice | NSD | Time per Image |
|-------|------|-----|----------------|
| LiteMedSAM | 61.70 | 65.47 | 13.7s |
| k-means OR threshold OR LiteMedSAM | 69.96 | 77.98 | 2.4s |

$256 \times 256$ to fit its input size. We believe that a crop-then-infer approach would be beneficial for MedSAM as it would only resize the crop instead of the whole image. This way, the bounding box would not be miniature and MedSAM can focus on the detail of the cropped local instance. However, we would need to perform a study regarding this in future work to confirm our hypothesis.

### 4.6   Mammography

**Efficiency Strategies:** Similarly to dermoscopy, MobileUNet achieves a 2x faster prediction than MedSAM.

Table 10: Results on MedSAM's training dataset for Mammography

| Model | Dice | NSD | Time per Image |
|-------|------|-----|----------------|
| LiteMedSAM | 79.15 | 82.09 | 0.86s |
| MobileUNet | 86.55 | 88.92 | 0.38s |

**How to improve MedSAM on Mammography?** We cannot make a statement on how to improve MedSAM in this domain as we only evaluate it on the training data. However, it seems that this domain is underrepresented in MedSAM's training data, which leads to a mammography-specialized Mobile-UNet outperforming MedSAM.

### 4.7   X-Ray, Endoscopy, and Fundus

**Efficiency Strategies:** We did not improve the efficiency on this task as we used MedSAM in our final submission.

**How to improve MedSAM on X-Ray, Endoscopy, and Fundus?** We have limited expertise in these domains and cannot offer any hypotheses on how to improve MedSAM.

### 4.8   Quantitative results on validation set

Table 11 shows that our approach outperforms (on average) the baseline (LiteMed-SAM). However, we also show in Tables 5-10 that we are able to significantly improve the efficiency of the baseline while sacrificing a negligible amount of performance.

Table 11: Quantitative evaluation results. Baseline: LiteMedSAM. Ablations were done on the subsampling factors for CT and MRI volumes.

| Target | Baseline | | Ablation $s = 2$ | | Ablation $s = 8$ | | Proposed $s = 3$ | |
|---|---|---|---|---|---|---|---|---|
| | DSC(%) | NSD(%) | DSC(%) | NSD(%) | DSC(%) | NSD (%) | DSC(%) | NSD (%) |
| CT | 92.35 | 95.09 | 91.88 | 94.35 | 89.81 | 92.42 | 91.59 | 94.18 |
| MR | 89.93 | 94.02 | 88.99 | 92.93 | 84.99 | 89.58 | 88.34 | 92.57 |
| PET | 55.23 | 29.29 | 66.80 | 49.42 | 66.80 | 49.42 | 66.80 | 49.42 |
| US | 94.78 | 96.81 | 94.78 | 96.82 | 94.78 | 96.82 | 94.78 | 96.82 |
| X-Ray | 75.83 | 80.39 | 75.87 | 80.44 | 75.87 | 80.44 | 75.87 | 80.44 |
| Dermoscopy | 92.47 | 93.85 | 92.63 | 94.22 | 92.63 | 94.22 | 92.63 | 94.22 |
| Endoscopy | 96.04 | 98.11 | 96.04 | 98.11 | 96.04 | 98.11 | 96.04 | 98.11 |
| Fundus | 94.81 | 96.41 | 94.82 | 96.42 | 94.82 | 96.42 | 94.82 | 96.42 |
| Microscopy | 61.70 | 65.47 | 69.96 | 77.98 | 69.96 | 77.98 | 69.96 | 77.98 |
| Average | 83.68 | 83.27 | 85.75 | 86.74 | 81.05 | 83.16 | 85.65 | 86.68 |

## 4.9   Segmentation efficiency results on validation set

The efficiency on a few samples from the validation set are listed in Table 12. Our optimization on the CT, MR, PET, microscopy, and dermoscopy modalities contributes to a much more efficient prediction time.

Table 12: Quantitative evaluation of segmentation efficiency in terms of running time (s) on the hardware specified in Table 2. Abl.: Ablation, * Grayscale image

| Case ID | Size | #Objects | Baseline | Abl. $s = 8$ | Ours $s = 3$ |
|---|---|---|---|---|---|
| 3DBox_CT_0566 | (287, 512, 512) | 6 | 376.4 | 62.1 | 109.7 |
| 3DBox_CT_0888 | (237, 512, 512) | 6 | 100.5 | 19.0 | 29.4 |
| 3DBox_CT_0860 | (246, 512, 512) | 1 | 17.7 | 4.6 | 5.8 |
| 3DBox_MR_0621 | (115, 400, 400) | 6 | 157.1 | 22.6 | 42.4 |
| 3DBox_MR_0121 | (64, 290, 320) | 6 | 99.9 | 13.6 | 27.3 |
| 3DBox_MR_0179 | (84, 512, 512) | 1 | 17.1 | 4.2 | 5.6 |
| 3DBox_PET_0001 | (264, 200, 200) | 1 | 12.1 | 0.19 | 0.19 |
| 2DBox_US_0525 | (256, 256, 3) | 1 | 2.0 | 2.0 | 2.0 |
| 2DBox_X-Ray_0053 | (320, 640, 3) | 34 | 2.9 | 2.9 | 2.9 |
| 2DBox_Dermoscopy_0003 | (3024, 4032, 3) | 1 | 2.2 | 1.3 | 1.3 |
| 2DBox_Endoscopy_0086 | (480, 560, 3) | 1 | 2.0 | 2.0 | 2.0 |
| 2DBox_Fundus_0003 | (2048, 2048, 3) | 1 | 2.0 | 2.0 | 2.0 |
| 2DBox_Microscope_0008 | (1536, 2040, 3) | 19 | 2.6 | 2.6 | 2.6 |
| 2DBox_Microscope_0016 | (1920, 2560, 3) | 241 | 12.9 | 12.9 | 12.9 |
| *2DBox_Microscope_0030 | (2304, 2304, 3) | 137 | 7.4 | 1.6 | 1.6 |
| *2DBox_Microscope_0040 | (944, 1266, 3) | 56 | 3.4 | 0.3 | 0.3 |

### 4.10   Qualitative results on validation set

We show some qualitative image examples for the predictions of our models on various modalities.

**3D slice-wise inference.** Fig. 5 demonstrates the effect of various subsampling rates $s$ on a CT prediction. While the prediction is much more efficient with a higher rate, it also reduces the level of details and even introduces interpolation artifacts. This is one of the reasons that led us to set $s = 3$ to a lower value.

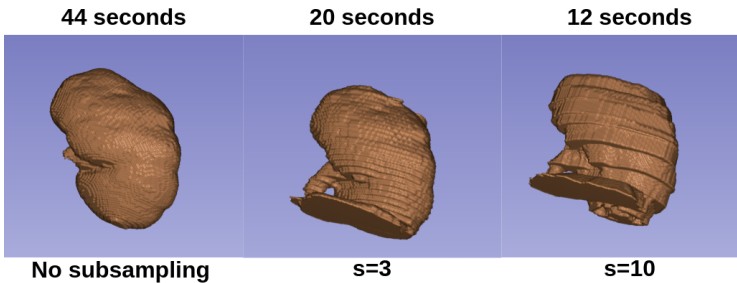

Fig. 5: Example of various subsampling factors $s$ for a kidney from the CT validation set. Higher factors introduce artifacts during the interpolation and smoothen out high-frequency details but improve the efficiency.

**Microscopy classical approaches.** Fig. 6 and Fig. 7 demonstrate failure cases of MedSAM in the microscopy domain. It seems that MedSAM struggles with small structures with ambiguous boundaries. In contrast, k-means clustering and mean thresholding perform quite well on these domains, landing a spot into our methodology for our final submission to the challenge.

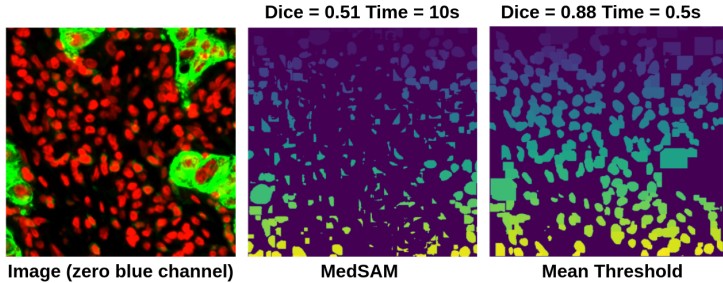

Fig. 6: Examples of predictions for two-channel microscopy images from the training set.

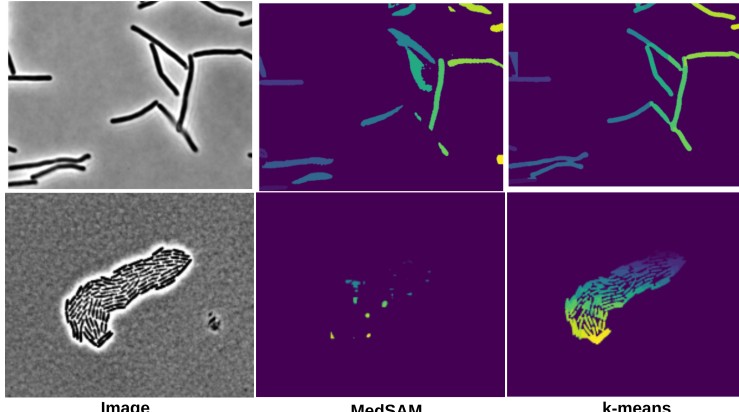

Fig. 7: Examples for grayscale microscopy images from the validation set.

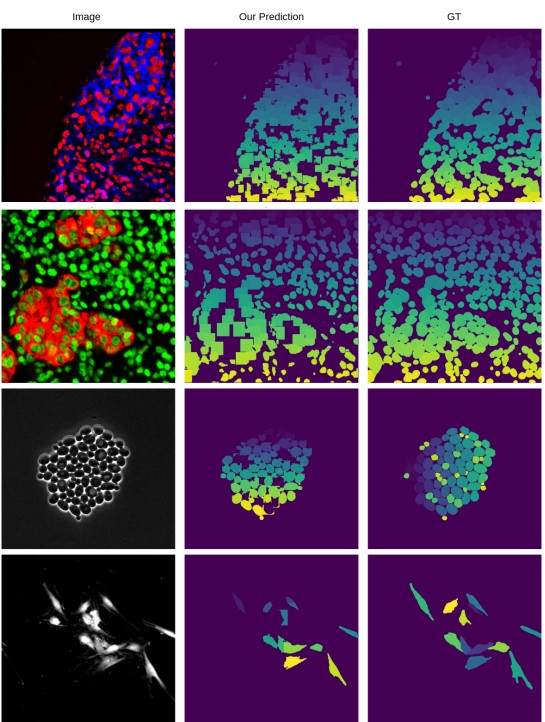

Fig. 8: Examples for failure cases of our microscopy predictions.

**Microscopy - Failure Cases.** Fig. 8 depicts examples of predictions on microscopy images. It can be seen that in some instances the k-means clustering and the mean threshold fill up the whole bounding box as the intensity within

the local instance is higher than the mean intensity in the global image or the pixels are homogenous. This is an inherent challenge in the microscopy domain.

**Dermoscopy, Endoscopy, Fundus, Mammography.** Fig. 9 shows examples for predictions on four imaging modalities. The labels are often quite coarse, which may lead to a low reported Dice and NSD although the segmentation mask on its own might be plausible.

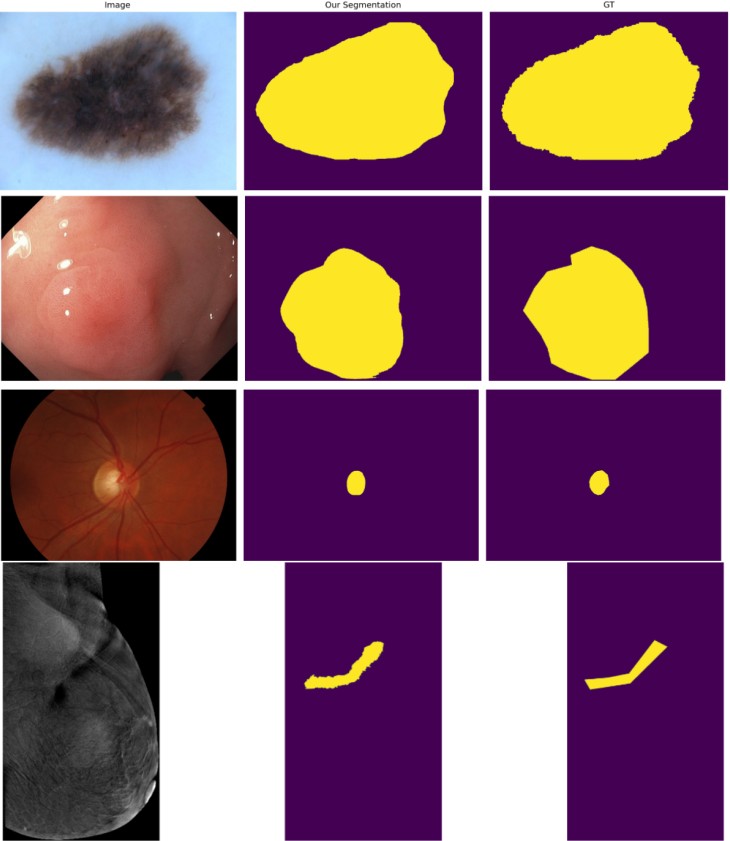

Fig. 9: Examples for Dermoscopy, Endoscopy, Fundus, and Mammography predictions

**PET and OCT threshold-based methods.** Fig. 10 shows examples of predictions on OCT and PET images. As confirmed in the quantitative results, thresholds produce plausible predictions for both modalities.

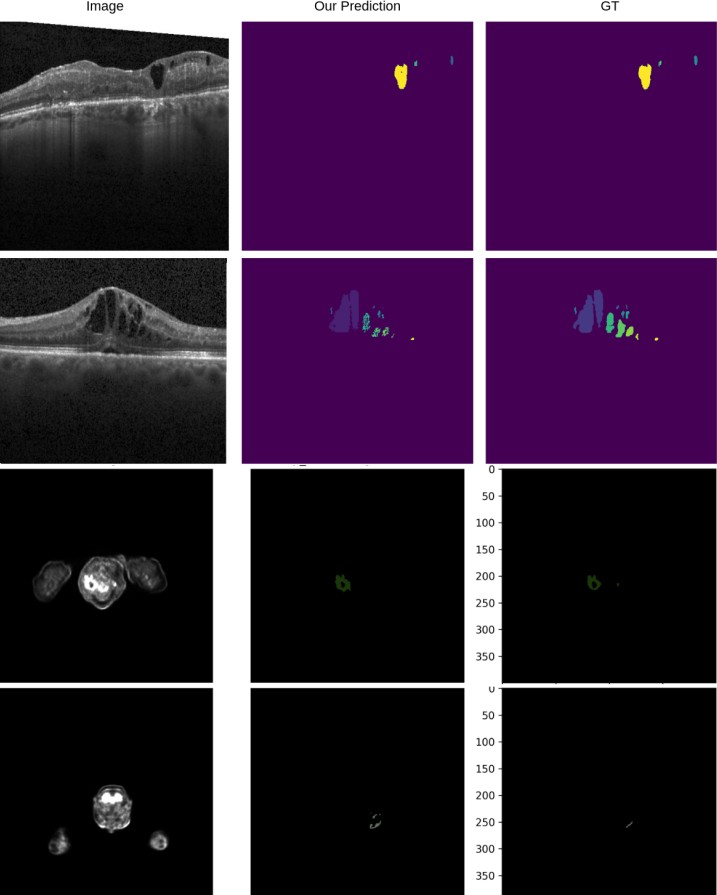

Fig. 10: Examples for OCT and PET predictions.

### 4.11    Results on final testing set

The results on the final test set are presented in Table 11. Our emphasis on classical methods is reflected in consistently efficient runtimes, consistently placing within the top six methods. However, while threshold-based approaches for OCT and PET demonstrate exceptional efficiency, they suffer from low generalizability. In fact, these methods perform the worst in terms of segmentation quality within the OCT domain. Interestingly, our submission achieved 1st, 2nd, and 4th place in the X-Ray, Endoscopy, and Fundus domains, respectively, in terms of segmentation performance. This success was achieved despite only reusing the baseline model (LiteMedSAM) and incorporating a minor post-processing step, as described in Section 2.9—specifically, omitting any predictions outside the bounding box and retaining the largest connected component and filling its holes. This demonstrates that LiteMedSAM already possesses a great gen-

eralization ability but could be substantially improved by adding some simple post-transformations. The results for CT and MR are unfortunately one of the worst, perhaps due to the introduced noise from our slice interpolation scheme described in 4.1. Our method is placed in the 6th place in the final leaderboard out of 23 methods.

Table 13: Results on the testing stage of the challenge. Worst rank is 23

| Modality | Dice (Rank) | NSD (Rank) | Runtime (Rank) |
|---|---|---|---|
| CT | 53.89 (18) | 54.27 (19) | 20.66 (5) |
| MR | 64.28 (18) | 62.61 (15) | 9.08 (5) |
| X-Ray | 85.79 (1) | 94.06 (1) | 5.10 (6) |
| Endoscopy | 94.41 (2) | 96.95 (2) | 4.58 (6) |
| Fundus | 86.47 (4) | 88.54 (4) | 4.59 (6) |
| Microscopy | 84.49 (13) | 86.30 (14) | 5.55 (6) |
| OCT | 65.52 (23) | 70.21 (23) | 1.11 (1) |
| PET | 74.18 (12) | 63.46 (15) | 1.07 (1) |
| US | 85.84 (12) | 90.13 (12) | 6.40 (6) |

### 4.12   Post-challenge Analysis

During the post-challenge phase, we have participated in the **Performance Booster** track without the use of any external datasets. Here, we describe how we have improved our methods.

**Changed Methodology** We made only a few adjustments to our methods, focusing on incremental improvements. For 3D-based modalities (CT and MRI, but not PET), we enhanced our implementation by processing bounding boxes in parallel using the `concurrent.futures` Python module, fixing the number of threads to 8. To ensure that the segmentation results matched those of sequential execution, we handled overlapping segmentations by prioritizing larger indices. Specifically, if two bounding boxes overlap in their predictions, the prediction from the last bounding box overwrites the earlier one in areas of conflict. We acknowledge that this is a naive solution to a complex problem, which requires more in-depth analysis. The only other method we changes is the OCT domain, as our original method was the last on the leaderboard. For this, we simply replaced the thresholding method with a trained MobileUNet on the original training data as described in Section 3.2.

**Results from Post-challenge Analysis** The changes in the code led to an improvement in the runtime for both the CT and MR domains in terms of runtime. For OCT, our method is three times slower than our previous thresholding approach, but it improves the Dice and NSD by over 10% each.

Table 14: Post-challenge changes in performance

| Modality | Dice | NSD | Runtime |
|---|---|---|---|
| CT | 53.89 | 54.27 | 19.07 |
| MR | 64.28 | 62.61 | 8.78 |
| OCT | 74.37 | 81.47 | 3.35 |

### 4.13  Limitation and future work

Our methodology has two main limitations: (1) it focuses on individual imaging modalities rather than proposing a unified framework like MedSAM; (2) classical non-deep learning approaches often depend on expert knowledge for optimal performance (e.g., for PET data, we assume that lesions are avid and bright in the image). However, we intentionally adopt this fragmented approach to identify the weaknesses in MedSAM and understand their underlying causes, aiming to improve it in future iterations. Our findings suggest that incorporating explicit assumptions about imaging modalities can serve as a robust signal, sometimes outperforming MedSAM in specific cases.

A promising future direction involves appending a "task adapter" to Med-SAM. This would function as a model capable of recognizing the task from the image, such as "fetal head segmentation in ultrasound," and providing this contextual information to MedSAM. This addition would enable MedSAM to produce more accurate and context-specific outputs, such as coarse and ellipsoid-shaped masks. In other words, informing the model about the specific segmentation task should enhance its adaptability and performance. This approach allows the integration of domain knowledge directly into the model, potentially adapting it to specific domains and tasks.

## 5  Conclusion

Our results indicate that classical approaches can outperform MedSAM on certain imaging modalities, both in segmentation accuracy and efficiency. Several factors contribute to this outcome, including the presence of coarse ground-truth labels in dermoscopy, the scarcity of large public datasets for mammography and PET, and the loss of detail during image resizing in microscopy images. Moreover, our findings highlight that integrating explicit task knowledge is crucial for surpassing MedSAM's performance. We propose that a task adapter, which provides information about the target structure and imaging modality, could enhance MedSAM's effectiveness in these challenging domains. Additionally, our slice interpolation experiments demonstrated that it is possible to improve MedSAM's efficiency without significantly compromising its segmentation performance.

**Acknowledgements**  We thank all the data owners for making the medical images publicly available and CodaLab [45] for hosting the challenge platform.

The present contribution is supported by the Helmholtz Association under the joint research school "HIDSS4Health – Helmholtz Information and Data Science School for Health. Parts of this work were performed on the HoreKa super-computer funded by the Ministry of Science, Research and the Arts Baden-Württemberg and by the Federal Ministry of Education and Research.

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

Table 15: Checklist Table. Please fill out this checklist table in the answer column.

| Requirements | Answer |
| --- | --- |
| A meaningful title | Yes |
| The number of authors (≤6) | 4 |
| Author affiliations and ORCID | Yes |
| Corresponding author email is presented | Yes |
| Validation scores are presented in the abstract | Yes |
| Introduction includes at least three parts: background, related work, and motivation | Yes |
| A pipeline/network figure is provided | Figure 1 |
| Pre-processing | Page 8 |
| Strategies to data augmentation | Page 10 |
| Strategies to improve model inference | Pages 3-8 |
| Post-processing | Page 9 |
| Environment setting table is provided | Tables 2 and 3 |
| Training protocol table is provided | Table 4 |
| Ablation study | Page 11, Table 5 |
| Efficiency evaluation results are provided | Table 12 |
| Visualized segmentation example is provided | Figures 5-10 |
| Limitation and future work are presented | Yes |
| Reference format is consistent. | Yes |
| Main text >= 8 pages (not include references and appendix) | Yes |