# OpenReview forum: "Taking a Step Back: Revisiting Classical Approaches for Efficient Interactive Segmentation of Medical Images"
_thecvf.com/CVPR/2024/Workshop/MedSAMonLaptop — CVPR24 MedSAMonLaptop_

### Official Review · Reviewer_DVUs · 2024-06-11
**Review: Great Paper and Analysis**

**Rating:** 9
**Confidence:** 5

**Review:**

The authors present a very interesting idea for this competition and accompany it with a well-written and exhaustive report.

I specifically want to highlight the following contributions: a detailed investigation of the training data, well-explained failure cases of MedSAM, a very insightful analysis of an approach for interpolation for 3D images, and a good presentation of the results.

There are no problems with this submission to the workshop, and I believe it fulfills all the requirements for the competition's write-up.

### Additional Feedback

* The reference to LiteMedSAM at the start is missing (but is provided later in Section 2.8.
* In some cases, like in Section 2.7., the authors mentioned that classical approaches did not work for a certain modality. I think it would still be highly insightful to share which approaches the authors tried but did not work!
* "For all 2D images, except the ones predicted by MedSAM, we keep the largest connected component and fill all the holes within it."
 This is very interesting, but it is not given any attention in the results. Did this make the performance much better? Do the authors, by chance, have an ablation study of this?
* In the provided code, the README could be updated to reflect your specific code changes and contributions.
* While reading the abstract and introduction, it seemed to me that your final submission would consist mostly of a traditional non-deep learning approach, but the opposite was the fact, as seen in Table 1 or Figure 1. I suggest reiterating the abstract and introduction to reflect this so that both do not oversell the results or confuse the reader.

---

### Official Review · Reviewer_UWTS · 2024-06-24
**great paper**

**Rating:** 7
**Confidence:** 5

**Review:**

The paper is really well-written and provides all the necessary details. I very much enjoy reading it!

It is also very impressive to see that classical methods can achieve promising results on some tasks in the deep learning era.
I only have two minor ments:

- There are lots of classical methods. Could you please add a paragraph to describe the motivations to determine the classical method for specific segmentation tasks?
- Could you please also comment on graph cut and active contours, which are also popular traditional methods?

---

### Official Review · Reviewer_54zy · 2024-06-24
**In-depth data analysis to choose best method for each modality**

**Rating:** 9
**Confidence:** 4

**Review:**

The paper comprehensively analyzes each modality of the dataset and decides on the method to use. It reviews failure cases of MedSAM and complements the method with classical approaches such as k-means or thresholding. For the 3D dataset, slice interpolation is incorporated to increase the efficiency.  Using the classical approaches with the supervision of the instance-level bounding boxes, it achieves good performance and efficiency.
I have minor constructive comments:
1. Highlighting the best metrics in the tables would be useful.
2. Commenting on how the coloring coding was done on images with multiple instances would be helpful.

---

### Decision · Program_Chairs · 2024-10-01

Accept